# Upper Normal Serum Creatinine Concentrations as a Predictor for Chronic Kidney Disease: Analysis of 14 Years’ Korean Genome and Epidemiology Study (KoGES)

**DOI:** 10.3390/jcm7110463

**Published:** 2018-11-21

**Authors:** Jong Hyun Jhee, Seun Deuk Hwang, Joon Ho Song, Seoung Woo Lee

**Affiliations:** Division of Nephrology and Hypertension, Department of Internal Medicine, Inha University, College of Medicine, Incheon 22332, Korea; jhjhee0119@inhauh.com (J.H.J.); lakisis79@hanmail.net (S.D.H.); jhsong@inha.ac.kr (J.H.S.)

**Keywords:** serum creatinine, estimated glomerular filtration rate, chronic kidney disease, proteinuria

## Abstract

Both serum creatinine (sCr) and estimated glomerular filtration rate (eGFR) have been used to assess kidney function in public health check-ups. However, when the sCr is within the normal levels but the eGFR is <60 mL/min/1.73 m^2^, a dilemma arises, as the patients might progress to chronic kidney disease (CKD) after several years. We aimed to evaluate the association between normal sCr and the risk of incident CKD in the general population. For this, 9445 subjects from the Korean Genome and Epidemiology Study, with normal sCr and eGFR of >60 mL/min/1.73 m^2^ were analyzed. The subjects were classified into quartiles based on sCr levels. The primary outcome was the development of eGFR <60 mL/min/1.73 m^2^ on two consecutive measures. During a mean follow-up of 8.4 ± 4.3 years, 779 (8.2%) subjects developed eGFR <60 mL/min/1.73 m^2^. The incidence of the development of eGFR <60 mL/min/1.73 m^2^ was higher in the higher quartiles than in the lowest quartile. In multivariable Cox analysis, the highest quartile was associated with an increased risk for the development of eGFR <60 mL/min/1.73 m^2^ (hazard ratio (HR), 4.71; 95% confidence interval (CI), 3.29–6.74 in females; HR, 12.77; 95% CI, 7.69–21.23 in males). In the receiver operating characteristic curve analysis, adding sCr to the traditional risk factors for CKD improved the accuracy of predicting the development of eGFR <60 mL/min/1.73 m^2^ (area under the curve, 0.83 vs. 0.80 in females and 0.85 vs. 0.78 in males), and the cutoff value of sCr was 0.75 mg/dL and 0.78 mg/dL in females and males. Cautious interpretation is necessary when sCr is within the normal range, considering that the upper normal range of sCr has a higher risk of CKD development.

## 1. Introduction

Chronic kidney disease (CKD) is increasingly recognized as a major global public health problem [1]. Patients with CKD are at an increased risk for cardiovascular disease (CVD), end-stage renal disease (ESRD), and all-cause mortality [2,3]. Both globally and in Korea, the incidence of CKD has been increasing over the last decade [4,5]. Hence, early identification of people with a high risk for CKD is crucial for primary prevention and to reduce the public health burden [6].

In Korea, the national health screening program (KNHSP) has been conducted every 2 years since 2002 for the early detection of chronic diseases and for public health promotion [7]. Initially, a urine dipstick test was performed for proteinuria and hematuria for the early detection of CKD, and if the results for proteinuria or hematuria were positive, blood urea nitrogen (BUN) and serum creatinine (sCr) concentrations were investigated [8,9]. In KNHSP, the presence of CKD is determined by either high sCr (>1.5 mg/dL) or low estimated glomerular filtration rate (eGFR <60 mL/min/1.73 m^2^), and the patient is notified of “suspected kidney dysfunction”. However, when both sCr and eGFR are used to screen for CKD in KNHCP, primary care physicians face a challenge in the diagnosis of CKD. There are four possible scenarios: (1) high sCr and low eGFR, (2) high sCr and normal eGFR, (3) normal sCr and low eGFR, and (4) normal sCr and normal eGFR. In Scenario 3 (normal sCr and low eGFR), it is difficult for the doctors to interpret whether the kidney function is normal because most guidelines recommend using eGFR, not sCr as the definition of CKD [10,11,12], and persons meeting Scenario 3 (normal sCr and low eGFR) might indeed progress to CKD. This might lead to a debate that (1) the upper normal limit of sCr should be reset to be equal to an eGFR of 60 mL/min/1.73 m^2^, and (2) sCr close to the upper normal limit might have a higher risk for CKD. In addition, patients whose eGFR is >60 mL/min/1.73 m^2^ are not aware of their kidney function and there is no management strategy for such cases.

However, there have been no studies evaluating sCr as a predictive marker for CKD in subjects who present within the normal range of eGFR. Therefore, this study was based on the hypothesis that sCr levels might have prognostic value for future renal function decline in subjects with normal eGFR at the initiation of the study.

## 2. Materials and Methods

### 2.1. Study Subjects

The Korean Genome and Epidemiology Study (KoGES) was a prospective community-based cohort study with repeated medical health check-up and various surveys, supported by the government. The detailed profile and methods on how the KoGES cohort was created have been previously described elsewhere [13]. Briefly, the study cohort consisted of 10,030 subjects aged 40–69 years who were residents of two cities in the Republic of Korea. Serial health examinations and surveys were performed biennially from 2001 to 2014. The results of this study were made public for research. From the study database, we selected subjects whose sCr levels were within the reference range based on previous reports [14,15], and whose eGFR was ≥60 mL/min/1.73 m^2^. We excluded those with underlying kidney disease at baseline, missing data, and missing follow-up sCr data. Individuals with underlying kidney disease were defined as those who were on treatment with diagnosis of CKD or taking medications due to kidney disease. A total of 9445 subjects were included in the final analysis. The female and male subjects were classified into quartiles according to the baseline sCr levels (Figure 1). All subjects voluntarily participated in the study, and provided informed consent. This study was performed in accordance with the Declaration of Helsinki and was approved by the institutional review board of Inha University Hospital (INHAUH 2018-06-007).

### 2.2. Anthropometric and Laboratory Data

All subjects underwent a comprehensive medical health examination and filled out questionnaires on health and lifestyle at the time of enrollment. Demographic and socioeconomic data, including age, sex, smoking status, alcohol intake, physical activity, and medical histories were obtained. Anthropometric parameters such as height and body weight were measured by skilled study workers following the standard methods. Body mass index (BMI) was calculated as weight divided by height squared (kg/m^2^). The body muscle mass was measured by multifrequency bioelectrical impedance analysis (InBody 3.0, Biospace, Seoul, Korea). Subjects with blood pressure (BP) ≥140/90 mmHg or taking anti-hypertensive agents were considered to be hypertensive. Patients with a fasting blood glucose level ≥126 mg/dL, glucose level ≥200 mg/dL after a 75 g oral glucose tolerance test, and hemoglobin A1c (HbA1c) ≥6.5% or those taking oral medication and/or insulin treatment for hyperglycemia were considered to have diabetes mellitus (DM). Subjects with a medical history of dyslipidemia or taking medications for lipid control were considered to have dyslipidemia. Cardiovascular diseases (CVDs) were defined as the composite of myocardial infarction, congestive heart failure, or coronary artery disease.

The following biochemical data were determined using fasting blood samples: concentrations of BUN, creatinine, albumin, glucose, total cholesterol, triglyceride, low-density lipoprotein-cholesterol (LDL-C), high-density lipoprotein-cholesterol (HDL-C), HbA1c, hemoglobin, and C-reactive protein (CRP). sCr levels were measured by the Jaffé method using ADVIA 1650 (Siemens, Tarrytown, NY, USA), which was traceable to an isotope dilution mass spectrometry reference method [16]. Urine samples were obtained in the morning after the first voiding and were subjected to the dipstick test using URISCAN Pro II (YD Diagnostics Corp., Seoul, Korea). Proteinuria was quantified as absent, trace, 1+, 2+, or 3+ based on a color scale. The presence of proteinuria was defined as the dipstick urine test showing greater than or equal to the 1+ level. eGFR was calculated using the CKD epidemiology collaboration (EPI) equation [17]. sCr level as well as other laboratory parameters were measured in two KoGES central laboratories. The intra- and inter-laboratory reliability of the serological parameters has been previously confirmed [13,18].

### 2.3. Study Outcomes

The primary end point was the development of eGFR <60 mL/min/1.73 m^2^ in two or more consecutive measurements during the follow-up period. In addition, the secondary end point was the development of proteinuria defined as ≥1+ level on a dipstick urine test and was evaluated in 8503 subjects who had no proteinuria at baseline.

### 2.4. Statistical Analysis

All statistical analyses were performed with IBM SPSS software for Windows version 23.0 (IBM Corporation, Armonk, NY, USA), SAS software version 9.2 (SAS Institute Inc., Cary, NC, USA), and R software version 3.3.1 (R Foundation for Statistical Computing, Vienna, Austria). Continuous variables were expressed as mean ± standard deviation, and categorical variables as absolute numbers with percentages. All data were tested for normality before the statistical analysis. The Kolmogorov-Smirnov test was performed to determine the normality of the distribution of the parameters. Comparisons between the groups were performed using analysis of variance or Student’s *t*-test for continuous variables with normal distribution, and the chi-square test or Fisher’s exact test for categorical variables. Data that did not show a normal distribution were presented as medians with an interquartile range and were compared using the Mann-Whitney U-test or Kruskal-Wallis test. Cumulative renal survival rates were estimated with Kaplan-Meier analysis and a log-rank test. Survival time was defined as the time interval between the baseline and the onset of outcome or the last follow-up. Patients who were lost to follow-up or had died were censored. Cox proportional hazards models were constructed to determine the independent predictive value of sCr in the development of eGFR <60 mL/min/1.73 m^2^. Variables that showed statistical significance in the univariable analyses or those considered to have a clinical significance were included in the multivariable models. All results were expressed as hazard ratio (HR) and 95% confidence interval (CI). Receiver operating characteristic (ROC) curves were plotted, and the area under the curve (AUC) was considered to indicate the predictive value of sCr in discriminating the development of eGFR <60 mL/min/1.73 m^2^. In addition, the best discrimination cutoff value for sCr levels was explored, which corresponds to the value where sensitivity plus specificity is maximized. Sensitivity analysis was performed with subgroups stratified age groups. For all analyses, a *p*-value < 0.05 was considered statistically significant.

## 3. Results

### 3.1. Baseline Characteristics

As shown in Figure 1, in all, 4954 females and 4491 males were analyzed in this study. The sCr ranges of each quartile were as follows: 0.4–0.6 mg/dL in Q1, 0.7 mg/dL in Q2, 0.8 mg/dL in Q3, and 0.9–1.1 mg/dL in Q4 among female subjects; 0.3–0.8 mg/dL in Q1, 0.9 mg/dL in Q2, 1.0 mg/dL in Q3, and 1.1–1.2 mg/dL in Q4 among male subjects. The baseline characteristics of the study subjects are shown in Table 1. The mean age was 52.1 ± 8.9 years, and 4491 (48.0%) subjects were male. The mean concentrations of sCr and eGFR at baseline were 0.81 ± 0.8 mg/dL and 94.3 ± 13.9 mL/min/1.73 m^2^, respectively. In the higher quartiles, females were older while males were younger. In both female and male subjects, BMI and muscle mass were larger in the higher quartiles. Smoking or alcohol status were not significantly different across the quartile groups. In male subjects, higher quartiles showed lower systolic blood pressure (SBP) and higher diastolic blood pressure. The prevalence of hypertension (HTN) was higher in the higher quartiles in both female and male subjects. The prevalence of DM was not different across the quartiles in females, while it was lower in the higher quartiles in males. However, the prevalence of dyslipidemia or CVD was not different between the quartiles in either female or male subjects. In laboratory tests, both female and male subjects showed higher baseline BUN, total cholesterol, and LDL-C levels, and lower eGFR levels in the higher quartiles. The prevalence of proteinuria was higher in the higher quartiles in females, whereas it was not different across the quartiles in males. In female subjects, higher quartiles showed lower baseline hemoglobin, HDL-C, CRP, and higher albumin levels. In the male subjects, higher quartiles showed higher baseline albumin, hemoglobin, and HDL-C levels, and lower HbA1c levels.

### 3.2. The Decline Rate of eGFR According to sCr Levels

We evaluated the decline rate of eGFR and the factors associated with this rate of decline (Appendix A). The median of decline rate of eGFR was 2.11 (interquartile range (IQR), 1.20–3.37) in females and 1.57 (IQR, 0.70–2.49) in males. We performed linear regression analysis to evaluate the significant clinical factors’ association with the decline rate of eGFR. In females, age, BMI, SBP, smoking status, history of HTN and DM, fasting plasma glucose, total cholesterol, and presence of proteinuria were significantly and positively associated with the decline rate, whereas alcohol status and serum albumin level showed negative correlations with the decline rate. In males, age, SBP, history of HTN, DM, and CVDs, fasting plasma glucose, total cholesterol, and presence of proteinuria were positively associated with the decline rate, whereas muscle mass, hemoglobin, serum albumin, and total cholesterol level showed negative relationships.

### 3.3. Risk of the Development of eGFR <60 mL/min/1.73 m^2^

During a mean follow-up of 8.4 ± 4.3 years, the development of eGFR <60 mL/min/1.73 m^2^ occurred in 779 (8.2%) subjects. The incidence of the development of eGFR <60 mL/min/1.73 m^2^ was significantly higher in those in the higher quartiles compared to those in the lowest quartile in both female and male subjects (4.6%, 11.0%, 15.4%, and 16.4% in Q1, Q2, Q3, and Q4 among female subjects, respectively; 1.9%, 5.6%, 8.8%, and 11.4% in Q1, Q2, Q3, and Q4 among male subjects, respectively, *p* for trend <0.001 in both females and males) (Table 2). Kaplan–Meier curves were plotted to investigate the effect of sCr levels on the development of eGFR <60 mL/min/1.73 m^2^ (Figure 2). In both female and male subjects, the higher quartiles showed a significantly higher risk for the development of eGFR <60 mL/min/1.73 m^2^ compared to the lowest quartile as the reference group (*p* < 0.001). Furthermore, univariable Cox analysis for the development of eGFR <60 mL/min/1.73 m^2^ with clinical parameters was performed (Appendix A). As a result, age, body mass index, systolic blood pressure, history of HTN and DM, hemoglobin, fasting plasma glucose, total cholesterol, and presence of proteinuria were significantly associated with increased risk of the development of eGFR <60 mL/min/1.73 m^2^, whereas alcohol status and serum albumin were related to decreased risk in female subjects. Moreover, age, muscle mass, body mass index, systolic blood pressure, history of HTN, DM, and CVDs, fasting plasma glucose, and presence of proteinuria were significantly associated with increased risk of the development of eGFR <60 mL/min/1.73 m^2^, whereas alcohol status, hemoglobin, and serum albumin were related to decreased risk in male subjects. Multivariable Cox analysis was performed to determine the independent predictive value of sCr levels on the development of eGFR <60 mL/min/1.73 m^2^. After adjustment for age, body muscle mass, BMI, SBP, smoking and alcohol status, history of HTN, DM, CVDs, hemoglobin, fasting plasma glucose, albumin, total cholesterol, CRP, and proteinuria, an increased risk for the development of eGFR <60 mL/min/1.73 m^2^ was observed in those in the higher quartiles compared to those in the lowest quartile in both female and male subjects (Table 2). In female subjects, the highest quartile showed a 4.71 (95% CI, 3.29–6.74) times higher risk of the development of eGFR <60 mL/min/1.73 m^2^ than the lowest quartile, while in males, the highest quartile revealed a 12.77 (95% CI, 7.69–21.23) times higher risk. A cubic spline plot was constructed with sCr level as a continuous variable to determine the association between sCr and the adjusted HRs for the development of eGFR <60 mL/min/1.73 m^2^. The plot revealed a non-linear association in both female (Figure 3A) and male subjects (Figure 3B). Hazard ratios (HRs) for the development of eGFR <60 mL/min/1.73 m^2^ increased as sCr levels increased. 

### 3.4. Risk of Proteinuria Development

Next, we evaluated the association between sCr levels and the development of proteinuria among 8503 subjects who had no proteinuria at baseline. During a mean follow-up of 8.4 ± 4.3 years, proteinuria developed in 1447 (15.3%) subjects. The incidence of proteinuria was significantly higher in highest quartile compared to the lowest quartile in both female and male subjects (9.1%, 10.5%, 13.3%, and 19.4% in Q1, Q2, Q3, and Q4, respectively, among female subjects; 11.7%, 19.0%, 26.3%, and 38.7% in Q1, Q2, Q3, and Q4 respectively among male subjects; *p* < 0.001 in both females and males). In multivariable Cox analysis after adjustment for age, body muscle mass, BMI, SBP, smoking and alcohol status, history of HTN, DM and CVDs, hemoglobin, fasting plasma glucose, albumin, total cholesterol, and CRP, the increased risk for proteinuria development was observed in highest quartile (HR, 4.87; 95% CI, 3.41–6.96 in female subjects; HR, 13.06; 95% CI, 7.86–21.69 in male subjects; Table 3).

### 3.5. Predictive Value of sCr Levels

In ROC curve analysis, sCr levels improved the accuracy of predicting the development of eGFR <60 mL/min/1.73 m^2^ when added to the traditional risk factors for CKD, including age, SBP, BMI, history of HTN, DM, CVDs, hemoglobin, fasting plasma glucose, albumin, and total cholesterol in both female and male subjects (AUC 0.80 vs. 0.83; *p* < 0.001 in female subjects; AUC 0.78 vs. 0.85; *p* < 0.001 in male subjects; Figure 4A,B). The cutoff value for sCr, with maximal sensitivity and specificity for prediction of the development of eGFR <60 mL/min/1.73 m^2^, was 0.75 mg/dL in female and 0.78 mg/dL in male subjects.

### 3.6. Sensitivity Analysis

First, as age-related decline in kidney function is inevitable, we further performed subgroup analysis stratified by different age groups: 40–50, 50–60, >60 years. As a result, higher sCr showed increased risk for the development of eGFR <60 mL/min/1.73 m^2^ across all subgroups (Appendix A).

## 4. Discussion

In this study, we demonstrated that the upper normal of sCr levels is predictive for the development of eGFR <60 mL/min/1.73 m^2^ as well as proteinuria in subjects with normal eGFR. A non-linear association was observed between sCr levels and the risk of the development of eGFR <60 mL/min/1.73 m^2^. Adding sCr levels to the traditional CKD risk factors, the predictive value for the development of eGFR <60 mL/min/1.73 m^2^ significantly improved. The levels of 0.75 and 0.78 mg/dL in female and male subjects showed maximal sensitivity and specificity for the prediction of the development of eGFR <60 mL/min/1.73 m^2^.

sCr is a typical endogenous filtration marker, and is a parameter for measuring eGFR in patients with CKD or ESRD [19,20]. Thus, there has been less concern regarding its prognostic value for clinical outcome. Recently, a few studies have suggested that sCr level itself might have a predictive role for decline in renal function, CVDs, or mortality in CKD patients, mainly due to its non-GFR determinant [21]. Bhavsar et al. [22] analyzed 865 African American subjects with hypertensive CKD, with a measured GFR (mGFR) of 20–65 mL/min/1.73 m^2^. They evaluated the association between the quartile of sCr as well as other endogenous filtration markers with the occurrence of ESRD. The study results showed that the highest quartile of sCr was significantly associated with a higher incidence and risk of progression to ESRD. These results were consistent after adjustment for mGFR, suggesting that the non-GFR determinant factors have an additional predictive role in kidney function. Tangri et al. [23] also performed an analysis using data from the Modification of Diet in Renal Disease (MDRD) study, which provided the baseline levels of filtration markers including sCr. They included 816 subjects with mGFR of 25–55 mL/min/1.73 m^2^ and examined associations between the filtration markers, kidney failure, and mortality. After adjustment for mGFR, higher levels of sCr were directly associated with an increased risk of kidney failure, whereas they were associated with a lower risk of mortality. Although these two studies demonstrated the role of sCr as a predictor for decline in kidney function, they were limited to patients with an already decreased GFR. One study evaluated the association between sCr and kidney function in nondiabetic CKD patients with a better preserved GFR, only excluding patients whose sCr was greater than 6 mg/dL [24]. The results were consistent with previous studies. Higher levels of sCr were significantly associated with a higher risk of kidney failure, defined as the doubling of sCr or progression to ESRD. However, the results of this study were also confined to CKD patients, and the size of the study cohort was relatively small. To date, no study has evaluated the association between sCr and kidney function among the general population. To the best of our knowledge, this study is the first to investigate the effect of sCr as a prognostic marker for CKD development in subjects with normal eGFR.

The plausible explanations for sCr as a predictive marker for CKD development are as follows. First, tubular secretion of creatinine is impaired early in the course of CKD development [25,26]. sCr levels are traditionally known to rise only after renal function is reduced by at least 50%. GFR of 50–60 mL/min/1.73 m^2^ is called a creatinine-blind range, in which sCr values fail to rise above the threshold [27,28]. This creatinine-blind range is explained by the compensatory mechanism of increased tubular secretion of creatinine in patients with mildly impaired renal filtration [29,30]. However, in subjects whose renal function is consistently declining, this compensatory ability reaches its maximum possible limit and the tubular secretion of creatinine is reduced, which consequently leads to CKD. Thus, early impairment of the tubular creatinine secretion can cause an elevation of sCr levels, and this elevated sCr level has a role in the prediction for CKD. Second, interestingly, the prevalence of HTN and proteinuria was higher compared to other ethnic groups [31]. In particular, female subjects showed much higher prevalence of HTN compared to males. It can be assumed that the higher prevalence of IgA nephropathy in Asians may have affected the higher proportion of HTN in this study. Furthermore, it is well-known that middle-aged to old women show a higher prevalence of HTN compared to men due to pregnancy-related elevation of blood pressure, menopausal effect, use of oral contraceptives, or hormonal replacement therapy [32,33,34,35]. In fact, the KoGES study is composed of middle-aged subjects ranging from 40 to 69, and this age-related increasing prevalence of HTN in women might also have been shown in this study. Notably, this higher prevalence of HTN and proteinuria in higher sCr groups indicates that subjects with normal but higher sCr levels already have early target organ damages and endothelial dysfunction compared to those with lower sCr levels, even though they are younger. Consequently, these early damages might have played a role in the loss of kidney function [36]. Additionally, the clinical parameters including age, BMI, SBP, history of HTN and DM, hemoglobin, fasting plasma glucose, and serum albumin were shown to associate with kidney function. Thus, it can be suggested that subjects with high-normal sCr and with pre-existing conditions, such clinical risk factors should be paid more attention in health care units [37,38]. Another explanation is the origin of the reference value for sCr. The upper reference limits have often been defined by the 95th percentile or mean plus 1.65 standard deviations in the random population, without consideration for the presence of subclinical disease [24]. From the US population data, 11% of individuals were found to have kidney disease or decreased kidney function, and 6% of the population had kidney disease with a GFR >60 mL/min/1.73 m^2^ [39]. Thus, many individuals referred to as having a “normal” creatinine concentration who are below the 95th percentile might have unidentified but impaired renal function. Accordingly, our study also found high-normal sCr levels to be associated with mildly reduced eGFR or proteinuria. Furthermore, there was a large difference of eGFR between Q1 and Q4 group. We assumed that the differences of eGFR between Q1 and Q4 were mainly due to the wide acceptable range of normal sCr concentration. Our study results suggested that the cut-off value for the prediction of CKD development was 0.75 mg/dL for females and 0.78 mg/dL for males. In light of these cut-off values, it is worthwhile to subdivide normal sCr values, which may aid in the risk stratification of kidney disease. For instance, HTN was previously subclassified with the labels normal, prehypertension, and HTN [40]. Likewise, sCr levels also need to be classified into normal, high-normal, and abnormal, and clinicians should pay particular attention to those with high-normal sCr as well as abnormal sCr. The strengths of this study are that it is the first study investigating the association between sCr levels with the risk of CKD development in subjects with normal eGFR. We performed the analysis using a well-designed and community-based cohort study with a long follow-up duration. We were able to examine these associations in an unadjusted model as well as in a model adjusted with muscle mass, which is a major non-GFR determinant and an important risk factor for the development of CKD. This allowed us to delineate the independent role of sCr levels in the prediction of CKD development. Thus, sCr can be applied as a clinical tool for risk stratification in the general population, especially those with normal eGFR.

This study has several limitations. First, by virtue of prospective study design, we could not measure the creatinine clearance or GFR directly. mGFR is the gold standard for the assessment of kidney function. Most recently, Dana V. Rizk et al. reported a novel and rapid technique for direct GFR measurement using visible fluorescent injectate [41]. However, some studies reported that mGFR has a much higher coefficient of variation than endogenous filtration markers, indicating that biological variations and measurement errors exist [42]. Steady-state serum levels of creatinine might reflect true GFR more accurately than mGFR. Second, we defined CKD based on eGFR as calculated by the CKD-EPI equation. There might be a discrepancy in eGFR based on how it is calculated. Furthermore, it can be obvious that higher sCr groups show lower eGFR levels. However, we pursued that an sCr value within the normal range can predict future kidney function decline and stratify the risk of incident CKD according to its level. In the clinical field, subjects with normal sCr level and normal eGFR level are considered to be healthy, and receive no further exams. For example, a subject with an eGFR of 70 mL/min/1.73 m^2^ and one with an eGFR of 100 mL/min/1.73 m^2^ are both treated as normal, and no further action is performed upon either of them. However, based on the results of our study, the risk for incident CKD was apparently different between subjects with eGFRs of 70 and 100 mL/min/1.73 m^2^. This indicates that subjects with higher sCr levels should take further care, even within normal sCr ranges. Additionally, to overcome the limitation of defining the primary outcome based on sCr, we defined secondary outcomes with incident proteinuria and the relative risk of CVDs, which are independent of sCr levels. The results showed that higher sCr was associated increased risk for incident proteinuria and the increasing trend of relative risk of CVDs (Appendix A). Finally, our results cannot be generalized because the measurement of sCr can vary from case to case. sCr levels are mainly affected by age, sex, race, and muscle mass. We adjusted for these factors, but the results were consistent. Nevertheless, the interpretation of our findings requires caution in different clinical settings. In view of these study limitations, further studies are warranted to validate the role of sCr as a predictive marker for CKD development.

## 5. Conclusions

In conclusion, a non-linear association was observed between sCr levels and the risk of CKD development. In addition, our study suggests cutoff values of sCr levels for prediction of the development of eGFR <60 mL/min/1.73 m^2^ as 0.75 mg/dL for females and 0.78 mg/dL for males. This study has clinical implications, in that close follow-up is needed in subjects with normal eGFR but with upper-normal sCr levels.

## Figures and Tables

**Figure 1 jcm-07-00463-f001:**
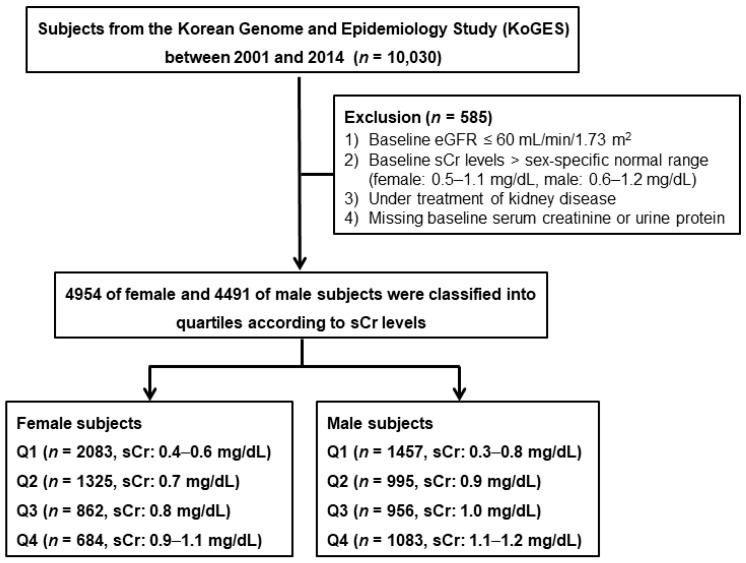
Study subjects. eGFR: estimated glomerular filtration rate; sCr: serum creatinine.

**Figure 2 jcm-07-00463-f002:**
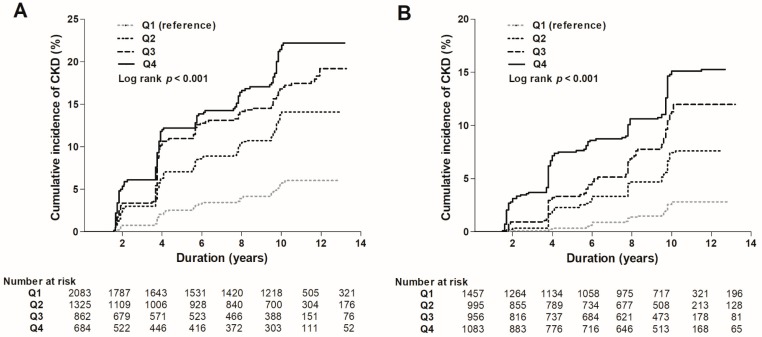
Kaplan-Meier curves for the development of eGFR <60 mL/min/1.73 m^2^ in female (**A**) and male (**B**) subjects.

**Figure 3 jcm-07-00463-f003:**
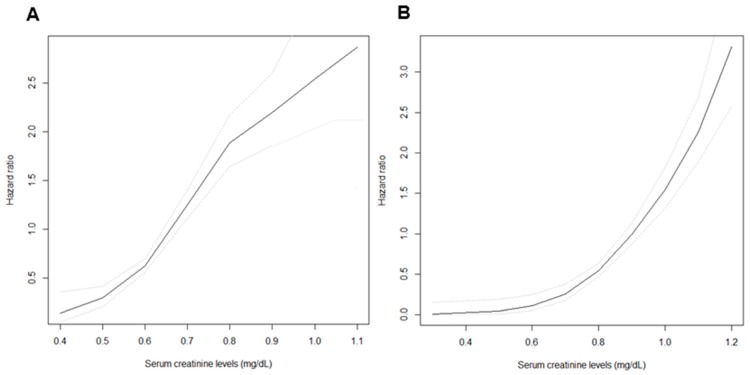
Cubic spline plots for the development of eGFR <60 mL/min/1.73 m^2^ according to sCr levels in female (**A**) and male (**B**) subjects. Black lines = hazard ratios, dotted lines = 95% confidence intervals.

**Figure 4 jcm-07-00463-f004:**
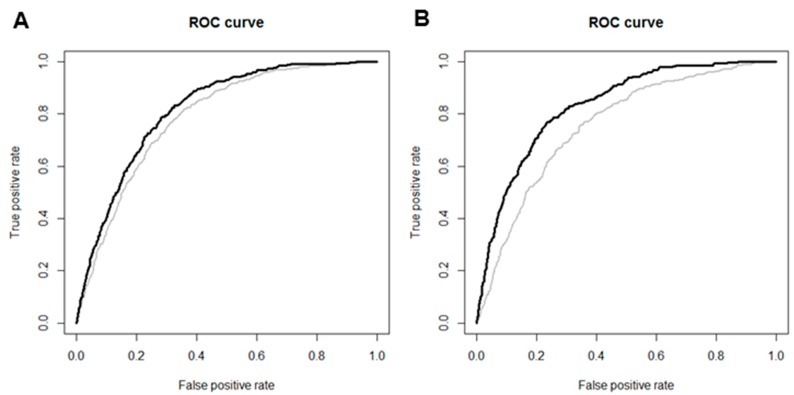
Receiver operating characteristic (ROC) curves for the development of eGFR <60 mL/min/1.73 m^2^ in female (**A**) and male (**B**) subjects. Areas under the curve (AUCs) were improved to 0.83 from 0.80 in female subjects and to 0.85 from 0.78 in male subjects when sCr levels were added to traditional risk factors for CKD. Gray lines represent ROC curves with traditional risk factors including age, BMI, SBP, history of HTN, DM, CVDs, hemoglobin, fasting plasma glucose, albumin, and total cholesterol. Black lines represent ROC curves with sCr levels added to traditional risk factors.

**Table 1 jcm-07-00463-t001:** Baseline characteristics according to quartiles of sCr levels in female and male subjects.

	Total (*n* = 9445)	Female (*n* = 4954)					Male (*n* = 4491)				
		Q1 (*n* = 2083)	Q2 (*n* = 1325)	Q3 (*n* = 862)	Q4 (*n* = 684)	*p*	Q1 (*n* = 1457)	Q2 (*n* = 995)	Q3 (*n* = 956)	Q4 (*n* = 1083)	*p*
Demographic data											
Age, years	52.1 ± 8.9	51.7 ± 8.8	52.3 ± 9.0	53.4 ± 9.1	53.2 ± 8.7	<0.001	53.8 ± 8.8	51.7 ± 8.5	51.1 ± 8.8	49.8 ± 8.5	<0.001
BMI, kg/m^2^	24.5 ± 3.1	24.7 ± 3.3	24.9 ± 3.2	25.0 ± 3.5	25.2 ± 3.3	0.025	23.5 ± 2.9	24.3 ± 2.8	24.3 ± 2.9	24.7 ± 2.7	<0.001
Body muscle mass, kg	43.5 ± 8.0	37.3 ± 4.2	37.7 ± 4.2	37.9 ± 4.4	38.4 ± 4.2	<0.001	48.0 ± 6.1	50.0 ± 5.7	50.5 ± 5.8	51.4 ± 5.9	<0.001
Smoking status, *n* (%)	3866 (41.5)	101 (5.0)	83 (6.4)	36 (4.3)	34 (5.0)	0.730	1186 (82.2)	780 (78.8)	778 (81.7)	868 (80.1)	0.396
Alcohol status, *n* (%)	5061 (54.1)	570 (27.7)	394 (30.1)	258 (30.3)	189 (27.8)	0.525	1198 (82.9)	790 (79.7)	789 (83.0)	873 (80.8)	0.396
SBP, mmHg	121.9 ± 18.6	121.8 ± 19.9	119.9 ± 19.4	121.5 ± 20.5	121.1 ± 19.1	0.062	123.9 ± 17.3	122.4 ± 16.9	123.3 ± 18.1	121.1 ± 16.2	<0.001
DBP, mmHg	80.5 ± 11.7	79.2 ± 12.3	78.4 ± 11.8	78.9 ± 12.2	79.4 ± 11.1	0.178	81.8 ± 10.8	81.7 ± 10.7	82.9 ± 11.5	82.8 ± 11.2	0.007
Comorbidities, *n* (%)											
Hypertension	1356 (14.4)	297 (14.3)	221 (16.7)	160 (18.6)	117 (17.1)	0.008	166 (11.4)	118 (11.9)	122 (12.8)	155 (14.3)	0.026
DM	611 (6.5)	124 (6.0)	60 (4.5)	49 (5.7)	44 (6.4)	0.775	132 (9.1)	68 (6.8)	66 (6.9)	68 (6.3)	0.009
Dyslipidemia	223 (2.4)	39 (1.9)	24 (1.8)	12 (1.4)	16 (2.3)	0.816	34 (2.3)	33 (3.3)	27 (2.8)	38 (3.5)	0.131
CVDs	148 (1.6)	25 (1.2)	21 (1.6)	12 (1.4)	10 (1.5)	0.570	22 (1.5)	14 (1.4)	17 (1.8)	27 (2.5)	0.062
Laboratory data											
Creatinine, mg/dL	0.81 ± 0.18	0.57 ± 0.04	0.70 ± 0.00	0.80 ± 0.00	0.92 ± 0.04	<0.001	0.75 ± 0.06	0.90 ± 0.00	1.00 ± 0.00	1.13 ± 0.05	<0.001
BUN, mg/dL	13.8 ± 3.6	11.9 ± 3.2	13.3 ± 3.0	14.2 ± 3.4	14.9 ± 3.9	<0.001	13.4 ± 3.4	14.6 ± 3.5	14.9 ± 3.8	15.5 ± 3.6	<0.001
eGFR, mL/min/1.73 m^2^	94.3 ± 13.9	107.0 ± 7.2	99.6 ± 6.2	84.1 ± 5.3	71.1 ± 5.4	<0.001	104.6 ± 7.5	98.2 ± 5.8	86.9 ± 5.3	75.2 ± 5.8	<0.001
Proteinuria (%)	702 (7.4)	118 (5.7)	93 (7.0)	61 (7.1)	57 (8.3)	0.012	123 (8.4)	82 (8.2)	81 (8.5)	87 (8.0)	0.771
Hemoglobin, g/dL	13.6 ± 1.6	12.5 ± 1.2	12.6 ± 1.1	12.6 ± 1.1	12.5 ± 1.1	0.007	14.5 ± 1.1	14.7 ± 1.1	14.8 ± 1.1	14.9 ± 1.1	<0.001
Albumin, g/dL	4.5 ± 0.2	4.4 ± 0.2	4.4 ± 0.2	4.4 ± 0.3	4.5 ± 0.2	<0.001	4.5 ± 0.3	4.6 ± 0.3	4.6 ± 0.2	4.7 ± 0.3	<0.001
Total cholesterol, mg/dL	198.0 ± 36.5	192.9 ± 35.2	199.6 ± 36.1	203.0 ± 37.1	208.9 ± 36.7	<0.001	186.3 ± 36.5	195.9 ± 33.6	203.2 ± 35.5	207.9 ± 36.2	<0.001
LDL-C, mg/dL	117.9 ± 34.4	114.5 ± 31.5	119.9 ± 32.1	125.2 ± 32.3	131.4 ± 34.1	<0.001	103.4 ± 35.2	113.4 ± 33.9	121.5 ± 35.6	128.2 ± 3.43	<0.001
HDL-C, mg/dL	49.6 ± 11.9	50.9 ± 11.9	52.1 ± 12.1	51.1 ± 11.5	50.1 ± 11.8	0.002	49.8 ± 13.0	48.6 ± 11.6	47.3 ± 10.7	45.5 ± 9.7	<0.001
Triglyceride, mg/dL	152.2 ± 109.9	137.5 ± 93.5	137.6 ± 90.8	133.6 ± 82.6	137.3 ± 94.9	0.717	165.7 ± 128.9	169.7 ± 136.9	172.0 ± 128.7	171.2 ± 104.7	0.587
Fasting glucose, mg/dL	92.3 ± 22.6	90.4 ± 21.4	89.9 ± 19.8	89.1 ± 15.2	91.3 ± 23.3	0.193	95.2 ± 27.1	93.3 ± 22.8	94.0 ± 19.2	95.8 ± 27.1	0.099
HbA1c, %	5.8 ± 0.9	5.8 ± 0.9	5.7 ± 0.8	5.8 ± 0.9	5.8 ± 0.9	0.279	5.9 ± 1.0	5.7 ± 0.8	5.7 ± 0.9	5.8 ± 0.9	0.005
CRP (IQR), mg/dL	0.14 (0.07–0.25)	0.14 (0.06–0.24)	0.14 (0.17–0.24)	0.14 (0.06–0.25)	0.12 (0.06–0.23)	0.042	0.15 (0.07–0.25)	0.14 (0.07–0.25)	0.15 (0.07–0.26)	0.14 (0.07–0.25)	0.804

Data are presented as mean ± standard deviation, median [interquartile range], or number (%). sCr: serum creatinine; BMI: body mass index; SBP: systolic blood pressure; DBP: diastolic blood pressure; DM: diabetes, mellitus; CVD: cardiovascular disease; BUN: blood urea nitrogen; eGFR: estimated glomerular filtration rate; LDL-C: low-density lipoprotein cholesterol; HDL-C: high-density lipoprotein cholesterol; CRP: C-reactive protein.

**Table 2 jcm-07-00463-t002:** Risk of the development of eGFR <60 mL/min/1.73 m^2^ according to quartiles of sCr levels in female and male subjects.

		Model 1		Model 2		Model 3		Model 4	
	Incidence of CKD (%) ^a^	HR (95% CI)	*p*	HR (95% CI)	*p*	HR (95% CI)	*p*	HR (95% CI)	*p*
Quartiles of sCr									
Female									
Q1 (*n* = 2083)	96 (4.6)	Reference		Reference		Reference		Reference	
Q2 (*n* = 1325)	146 (11.0)	2.50 (1.93–3.24)	<0.001	2.59 (1.90–3.53)	<0.001	2.62 (1.91–3.59)	<0.001	2.57 (1.83–3.60)	<0.001
Q3 (*n* = 862)	133 (15.4)	3.89 (2.99–5.06)	<0.001	3.51 (2.56–4.81)	<0.001	3.56 (2.58–4.92)	<0.001	3.79 (1.83–5.35)	<0.001
Q4 (*n* = 684)	112 (16.4)	4.20 (3.20–5.51)	<0.001	4.21 (3.04–5.84)	<0.001	4.19 (3.00–5.85)	<0.001	4.71 (3.29–6.74)	<0.001
Male									
Q1 (*n* = 1457)	28 (1.9)	Reference		Reference		Reference		Reference	
Q2 (*n* = 995)	56 (5.6)	2.88 (1.83–4.53)	<0.001	3.18 (1.90–5.34)	<0.001	3.08 (1.82–5.20)	<0.001	3.30 (1.90–5.74)	<0.001
Q3 (*n* = 956)	84 (8.8)	4.64 (3.03–7.12)	<0.001	5.79 (3.57–9.39)	<0.001	6.04 (3.72–9.81)	<0.001	6.75 (4.03–11.30)	<0.001
Q4 (*n* = 1083)	124 (11.4)	6.43 (4.27–9.69)	<0.001	10.31 (6.49–16.38)	<0.001	11.21 (7.02–17.90)	<0.001	12.77 (7.69–21.23)	<0.001

^a^*p* for trend <0.001. Model 1: Unadjusted model. Model 2: Adjusted for age and muscle mass. Model 3: Adjusted for Model 2 + BMI, SBP, smoking and alcohol status, history of HTN, DM, and CVDs. Model 4: Adjusted for Model 3 + hemoglobin, fasting plasma glucose, serum albumin, total cholesterol, CRP, and proteinuria. CKD: chronic kidney disease; HR: hazard ratio; CI: confidence interval.

**Table 3 jcm-07-00463-t003:** Risk of incident proteinuria development according to quartiles of sCr levels in female and male subjects.

		Model 1		Model 2		Model 3		Model 4	
	Incidence of proteinuria (%) ^a^	HR (95% CI)	*p*	HR (95% CI)	*p*	HR (95% CI)	*p*	HR (95% CI)	*p*
Quartiles of sCr									
Female									
Q1 (*n* = 2083)	174 (9.1)	Reference		Reference		Reference		Reference	
Q2 (*n* = 1325)	125 (10.5)	2.50 (1.93–3.24)	<0.001	2.59 (1.90–3.53)	<0.001	2.59 (1.89–3.55)	<0.001	2.60 (1.85–3.64)	<0.001
Q3 (*n* = 862)	104 (13.3)	3.89 (2.99–5.06)	<0.001	3.51 (2.56–4.81)	<0.001	3.56 (2.58–4.91)	<0.001	3.88 (2.76–5.47)	<0.001
Q4 (*n* = 684)	119 (19.4)	4.20 (3.20–5.51)	<0.001	4.21 (3.04–5.84)	<0.001	4.23 (3.03–5.91)	<0.001	4.87 (3.41–6.96)	<0.001
Male									
Q1 (*n* = 1457)	150 (11.7)	Reference		Reference		Reference		Reference	
Q2 (*n* = 995)	168 (19.0)	2.88 (1.83–4.53)	<0.001	3.19 (1.90–5.34)	<0.001	3.08 (1.82–5.20)	<0.001	3.31 (1.90–5.75)	<0.001
Q3 (*n* = 956)	225 (26.3)	4.64 (3.03–7.12)	<0.001	5.79 (3.57–9.39)	<0.001	6.04 (3.72–9.81)	<0.001	6.79 (4.05–11.37)	<0.001
Q4 (*n* = 1083)	382 (38.7)	6.43 (4.27–9.69)	<0.001	10.31 (6.49–16.38)	<0.001	11.21 (7.02–17.90)	<0.001	13.06 (7.86–21.69)	<0.001

^a^*p* for trend <0.001. Model 1: Unadjusted model. Model 2: Adjusted for age and muscle mass. Model 3: Adjusted for Model 2 + BMI, SBP, smoking and alcohol status, history of HTN, DM, and CVDs. Model 4: Adjusted for Model 3 + hemoglobin, fasting plasma glucose, serum albumin, total cholesterol, and CRP.

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
