# Peer review of "Upper Normal Serum Creatinine Concentrations as a Predictor for Chronic Kidney Disease: Analysis of 14 Years’ Korean Genome and Epidemiology Study (KoGES)"

_jcm, 2018, doi:10.3390/jcm7110463_

Reviewer 1 Report

I have two suggestions for the discussion:
The authors mention as a limitation of their study that they were unable to perform GFR measurement at the bedside. In fact, a method to do exactly this was recently published and should be discussed. ("A Novel Method for Rapid Bedside Measurement of
GFR" Dana V. Rizk,1 Daniel Meier,2 Ruben M. Sandoval,2,3 Teresa Chacana,1 Erinn S. Reilly,2
Jesse C. Seegmiller,4 Emmanuel DeNoia,5 James S. Strickland,2 Joseph Muldoon,2 and
Bruce A. Molitoris. J Am Soc Nephrol 29, 2018)

Secondly, the authors should reflect on the specifics of their cohort. Noteworthy, while proteinuria and hypertension are usually associated in a feed-forward loop and therefore increase together, this is only true in the female patients of this cohort but not in the males.
Could this be due to the incidence of IgA nephropathy, which is presumably much higher in this cohort as compared with White or Black patient cohorts? The authors should discuss this point.

Author Response

Responses to reviewers’ comments

# Referee 1

Comment 1: I have two suggestions for the discussion: The authors mention as a limitation of their study that they were unable to perform GFR measurement at the bedside. In fact, a method to do exactly this was recently published and should be discussed. ("A Novel Method for Rapid Bedside Measurement of GFR" Dana V. Rizk,1 Daniel Meier,2 Ruben M. Sandoval,2,3 Teresa Chacana,1 Erinn S. Reilly,2 Jesse C. Seegmiller,4 Emmanuel DeNoia,5 James S. Strickland,2 Joseph Muldoon,2 and Bruce A. Molitoris. J Am Soc Nephrol 29, 2018)

Answer: The authors thank the reviewer for this comment. We completely agree with the novel and rapid method of GFR measurement using visible fluorescent injectate (VFI) which is demonstrated by Dana V. Rizk et al. It is useful and reliable technique for assessing measured GFR. Thus, we added the mentioned reference in the Discussion section of the revised manuscript.

 Comment 2: Secondly, the authors should reflect on the specifics of their cohort. Noteworthy, while proteinuria and hypertension are usually associated in a feed-forward loop and therefore increase together, this is only true in the female patients of this cohort but not in the males. Could this be due to the incidence of IgA nephropathy, which is presumably much higher in this cohort as compared with White or Black patient cohorts? The authors should discuss this point.

Answer: The authors appreciate and completely agree with the reviewer’s comment. In South Korea, IgA nephropathy is most common primary glomerulonephritis and the prevalence reaches 27.5% among those [1]. Furthermore, it is well known that the prevalence of IgA nephropathy is much higher in Asian compared to other races [2]. This can be assumed for the reason why the cohort subjects showed such higher prevalence of hypertension and proteinuria  [3]. However, the KoGES cohort study does not offer the information about prevalence of IgA nephropathy and we could not have discussed this in the original manuscript, we mentioned above presumption in the revised manuscript.

Several studies reported that middle to old women show higher prevalence of hypertension compared to men  [4-6]. This phenomenon is explained by pregnancy related elevation of blood pressure, menopausal effect, use of oral contraceptives or hormonal replacement therapy [7-10]. In fact, the ‘KoGES study’ is composed of middle-aged subjects ranging from 40 to 69. We assumed that this age-related increasing prevalence of hypertension in women was also shown in our study. We depicted this in the Discussion section of the revised manuscript.

Reference:

1.          Lee, H.S.; Koh, H.I.; Lee, H.B.; Park, H.C. Iga nephropathy in korea: A morphological and clinical study. Clinical nephrology 1987, 27, 131-140.

2.          Barbour, S.J.; Cattran, D.C.; Kim, S.J.; Levin, A.; Wald, R.; Hladunewich, M.A.; Reich, H.N. Individuals of pacific asian origin with iga nephropathy have an increased risk of progression to end-stage renal disease. Kidney international 2013, 84, 1017-1024.

3.          Barbour, S.J.; Reich, H.N. Risk stratification of patients with iga nephropathy. American journal of kidney diseases : the official journal of the National Kidney Foundation 2012, 59, 865-873.

4.          August, P. Hypertension in women. Advances in chronic kidney disease 2013, 20, 396-401.

5.          Lloyd-Jones, D.M.; Evans, J.C.; Levy, D. Hypertension in adults across the age spectrum: Current outcomes and control in the community. Jama 2005, 294, 466-472.

6.          Pimenta, E. Hypertension in women. Hypertension research : official journal of the Japanese Society of Hypertension 2012, 35, 148-152.

7.          Coylewright, M.; Reckelhoff, J.F.; Ouyang, P. Menopause and hypertension: An age-old debate. Hypertension (Dallas, Tex. : 1979) 2008, 51, 952-959.

8.          Lindheimer, M.D.; Taler, S.J.; Cunningham, F.G. Hypertension in pregnancy. Journal of the American Society of Hypertension : JASH 2008, 2, 484-494.

9.          Chasan-Taber, L.; Willett, W.C.; Manson, J.E.; Spiegelman, D.; Hunter, D.J.; Curhan, G.; Colditz, G.A.; Stampfer, M.J. Prospective study of oral contraceptives and hypertension among women in the united states. Circulation 1996, 94, 483-489.

10.        Wassertheil-Smoller, S.; Anderson, G.; Psaty, B.M.; Black, H.R.; Manson, J.; Wong, N.; Francis, J.; Grimm, R.; Kotchen, T.; Langer, R. et al. Hypertension and its treatment in postmenopausal women: Baseline data from the women's health initiative. Hypertension (Dallas, Tex. : 1979) 2000, 36, 780-789.

Reviewer 2 Report

In this manuscript, authors demonstrated that individuals with higher serum creatinine within the normal range had higher risk of incident chronic kidney disease (CKD) even after adjustment for demographic and clinical/laboratory factors, and cut-off value for predicting CKD was 0.75 and 0.78 mg/dl in female and male, respectively, using a large community-based cohort in the Republic of Korea. The study seems well designed, and statistical analyses appear to be appropriately performed. However, there are some concerns in this study. The reviewer’s comments are described as following.

Major comments:

Authors suggested that higher serum creatinine was a risk factor for incident CKD. However, this attitude could not be accepted, because serum creatinine is a scale for renal function and CKD is defined by serum creatinine-based estimated GFR level. It is the same with that high blood pressure is a risk factor for hypertension. The outcomes should be specific conditions that is defined by not only serum creatinine, such as CKD-related cardiovascular events or ESRD.

As authors described in Discussion, GFR should be measured as inulin or iothalamate clearance. If renal function would be defined by measured GFR but not serum creatinine-based estimated GFR, serum creatinine might be a risk factor for CKD in similar studies.

In this study, middle-aged or older subjects were followed for approximately eight years. Nevertheless, age-related decline in renal function was not considered. As demonstrated in previous reports such as Baba M et al. Plos One 10(6);e0129036, 2015, eGFR should be decreased over time even at younger and healthy individuals. Therefore, it is obvious that subjects with higher serum creatinine are likely to reach lower eGFR level in future.

Since rate of decline in renal function (ml/min/year) differs substantially between individuals, baseline serum creatinine level could not predict the rate of decline in future. Therefore, the reviewer strongly recommends that factors affected decline in renal function, rather than baseline serum creatinine, should be analyzed in this cohort of subjects with apparently normal renal function. In addition, the rate of decline during the study period should be evaluated.

Minor comments:

Authors should explain how they excluded individuals with underlying kidney diseases.

Were serum creatinine and other parameters measured in a central laboratory or each institute?

Author Response

Responses to reviewers’ comments

# Referee 2

Comments to the Authors: In this manuscript, authors demonstrated that individuals with higher serum creatinine within the normal range had higher risk of incident chronic kidney disease (CKD) even after adjustment for demographic and clinical/laboratory factors, and cut-off value for predicting CKD was 0.75 and 0.78 mg/dl in female and male, respectively, using a large community-based cohort in the Republic of Korea. The study seems well designed, and statistical analyses appear to be appropriately performed. However, there are some concerns in this study. The reviewer’s comments are described as following.

Answer: The authors thank the reviewer for the thoughtful comments. The suggestions further improve the quality and integrity of the manuscript. Detailed point-by-point responses to comments are listed below.

 Comment 1: Authors suggested that higher serum creatinine was a risk factor for incident CKD. However, this attitude could not be accepted, because serum creatinine is a scale for renal function and CKD is defined by serum creatinine-based estimated GFR level. It is the same with that high blood pressure is a risk factor for hypertension. The outcomes should be specific conditions that is defined by not only serum creatinine, such as CKD-related cardiovascular events or ESRD.

Answer: The authors sincerely appreciate and agree with the reviewer’s comment. Because eGFR was calculated by sCr based equation (eGFR-EPI), it is obvious that higher sCr groups showed lower eGFR levels. However, we pursued that sCr value within normal range can predict future kidney function decline and stratify the risk for incident CKD according to its level. Furthermore, in public health medicine, the subject with normal sCr level as well as normal eGFR level is considered to be healthy and receives no further exams. The problem is that this decision is performed by primary care physicians who are not familiar to the significance of sCr concentration or eGFR. Even nephrologists also have a dilemma for the management of patients whose sCrs were near upper normal limit of sCr because there were no guidelines for this condition. For example, subjects with eGFR 70 or those with 100 mL/min/1.73 m2 are treated as same as normal, and no further action is performed upon both of them. However, from the results of our study, the risk for incident CKD was apparently different between subjects with eGFR 70 and 100 mL/min/1.73 m2. This indicates that subjects with higher sCr levels should take further cares even in normal sCr ranges.

In addition, the authors agree with the reviewer’s comment that “the outcomes should be specific conditions which is defined by not only sCr, such as CKD-related cardiovascular events or ESRD.” Thereby we used incident proteinuria as secondary endpoint. Additionally, we evaluated the relative risk for the cardiovascular events according to sCr quartiles. Although there were weak statistical significances, the risks of cardiovascular disease were tended to be higher in higher sCr groups, especially in male subjects (Supplemental Table 5). Regretfully, the study data, KoGES study, do not offer about ESRD events and we could not have investigated the association with the risk of ESRD. We discussed this in the Discussion of the manuscript and further augmented in the revised manuscript.

 Comment 2: As authors described in Discussion, GFR should be measured as inulin or iothalamate clearance. If renal function would be defined by measured GFR but not serum creatinine-based estimated GFR, serum creatinine might be a risk factor for CKD in similar studies.

Answer: The authors sincerely appreciate reviewer’s comment. We agree with the reviewer’s comment, but the direct measurement of GFR was impossible because of the prospective study design. However, the primary hypothesis for the present study was that a higher baseline sCr level even within the normal range may be a novel predictor for future kidney function deterioration. Therefore, management strategy should be concerned on those subjects. Nevertheless, it would be worthwhile to explore the association between directly measured GFR and sCr levels in the future study.

Nevertheless, to overcome this study limitation, we further performed the main analysis with eGFR calculated by Modification of Diet in Renal Disease (MDRD) study equation  [1]. Previous studies suggested that MDRD study equation is well correlated with direct measured GFR by inulin [2]. The results were equal to main analysis by eGFR-EPI equation that higher sCr quartile groups were associated with increased risk for incident CKD in both female and male subjects. We added this in Supplemental Table 4 and described in the Results section with subtitled sensitivity analysis and Discussion section of the revised manuscript.

Reference:

1.          Levey, A.S.; Coresh, J.; Greene, T.; Stevens, L.A.; Zhang, Y.L.; Hendriksen, S.; Kusek, J.W.; Van Lente, F. Using standardized serum creatinine values in the modification of diet in renal disease study equation for estimating glomerular filtration rate. Annals of internal medicine 2006, 145, 247-254.

2.          Oh, Y.J.; Cha, R.H.; Lee, S.H.; Yu, K.S.; Kim, S.E.; Kim, H.; Kim, Y.S. Validation of the korean coefficient for the modification of diet in renal disease study equation. The Korean journal of internal medicine 2016, 31, 344-356.

 Comment 3: In this study, middle-aged or older subjects were followed for approximately eight years. Nevertheless, age-related decline in renal function was not considered. As demonstrated in previous reports such as Baba M et al. Plos One 10(6);e0129036, 2015, eGFR should be decreased over time even at younger and healthy individuals. Therefore, it is obvious that subjects with higher serum creatinine are likely to reach lower eGFR level in future. Since rate of decline in renal function (ml/min/year) differs substantially between individuals, baseline serum creatinine level could not predict the rate of decline in future. Therefore, the reviewer strongly recommends that factors affected decline in renal function, rather than baseline serum creatinine, should be analyzed in this cohort of subjects with apparently normal renal function. In addition, the rate of decline during the study period should be evaluated.

Answer: The authors appreciate reviewer’s comment. We totally in accordance with the reviewer’s comment that there is age-related decline in renal function and eGFR decreases over time event at younger and healthy adults. Thus, we further performed subgroup analysis stratified by age group; 40-50, 50-60, and >60 years. As a result, higher sCr showed increased risk for incident CKD across all subgroups (Supplemental Table 3). We added this in the Result and Discussion section in the revised manuscript.

As recommended, we evaluated the decline rate of eGFR and the factors associated with this rate of decline (Supplemental Table 1). The rate of decline in eGFR over time was determined using the least squares linear regression of eGFR over time; this was calculated from serial sCr measured during the study period for each participant. The slope was expressed as the regression coefficient (mL/min/1.73 m2/year). The median of decline rate of eGFR was 2.11 [interquartile range (IQR), 1.20-3.37] in female and 1.57 [IQR, 0.70-2.49] in male. We performed linear regression analysis to evaluate the significant clinical factors association with the decline rate of eGFR. In female, age, BMI, SBP, smoking status, history of hypertension and diabetes, fasting plasma glucose, total cholesterol and presence of proteinuria were significantly and positively associated with the decline rate, whereas alcohol status and serum albumin level showed negative correlation with the decline rate. In male, age, SBP, history of hypertension, diabetes, and CVDs, fasting plasma glucose, total cholesterol, and presence of proteinuria were positively associated with the decline rate, whereas muscle mass, hemoglobin, serum albumin, and total cholesterol level showed negative relationship. In lights of these results, subjects with high-normal sCr and with preexisting such clinical risk factors should be paid more attention in the health care units  [3,4]. We added this in Result section in the revised manuscript.

Reference:

3.          Tangri, N.; Stevens, L.A.; Griffith, J.; Tighiouart, H.; Djurdjev, O.; Naimark, D.; Levin, A.; Levey, A.S. A predictive model for progression of chronic kidney disease to kidney failure. Jama 2011, 305, 1553-1559.

4.          Tangri, N.; Grams, M.E.; Levey, A.S.; Coresh, J.; Appel, L.J.; Astor, B.C.; Chodick, G.; Collins, A.J.; Djurdjev, O.; Elley, C.R. et al. Multinational assessment of accuracy of equations for predicting risk of kidney failure: A meta-analysis. Jama 2016, 315, 164-174.

Comment 1: Authors should explain how they excluded individuals with underlying kidney diseases.

Answer: “Underlying kidney disease’ was defined as those who were on treatment with diagnosis of CKD or taking medications due to kidney disease. This definition is originally constructed from the KoGES cohort study. We described this in Methods section in the revised manuscript.

Comment 2: Were serum creatinine and other parameters measured in a central laboratory or each institute?

Answer: Serum creatinine level as well as other laboratory parameters were measured in two KoGES central laboratories. The intra- and inter-laboratory reliability of the serological parameters has been previously confirmed  [5,6]. We added this in Methods section.

Reference:

5.          Kim Y, Han BG: Cohort Profile: The Korean Genome and Epidemiology Study (KoGES) Consortium. Int J Epidemiol 46(2): e20, 2017.

6.          Yang JJ, Yang JH, Kim J, Cho LY, Park B, Ma SH, Song SH, Min WK, Kim SS, Park MS, Park SK: Reliability of quadruplicated serological parameters in the korean genome and epidemiology study. Epidemiol Health 33: e2011004, 2011.

Reviewer 3 Report

The article was written about renal outcome according to serum creatinine levels in patients with normal eGFR. Even with normal kidney function, serum creatinine levels predicted the renal outcome. The present study was valuable in the points of large scale and long follow-up period fashion. The idea about the study was good in term that eGFR was not always an accurate marker in every CKD patient. However, there are some problematic points.

 1.          

There were significant differences of eGFR and blood urea nitrogen between the four groups. Especially, the differences of eGFR between the Q1 and the Q4 were more than 30 ml/min/1.73m2. I considered that the differences were strongly associated with baseline kidney function. Therefore, it was obvious that serum creatine level was a predictor.

2.          

Why did author add eGFR or BUN to model in analysis for outcomes? The author should conduct analysis stratified by serum creatinine levels.

3.          

From previous reports, some variables such as comorbid of diabetes and hypertension are related to renal outcomes including halving of GFR. In the present study, what were the valuables associated with outcomes?

4.          

The prevalence of diabetes was higher in the Q1 group, on the other hand, that of  hypertension was lower in the Q1 group. How did the author consider the reason?

5.          

The author dealt serum creatine with marker of filtration. If so, creatinine clearance should be added to analysis.

Author Response

Responses to reviewers’ comments

# Referee 3

Comments to the Authors: The article was written about renal outcome according to serum creatinine levels in patients with normal eGFR. Even with normal kidney function, serum creatinine levels predicted the renal outcome. The present study was valuable in the points of large scale and long follow-up period fashion. The idea about the study was good in term that eGFR was not always an accurate marker in every CKD patient. However, there are some problematic points.

Answer: The authors appreciate the reviewer for the thoughtful comments. The suggestions further improve the quality and integrity of the manuscript. Detailed point-by-point responses to comments are listed below.

Comment 1: There were significant differences of eGFR and blood urea nitrogen between the four groups. Especially, the differences of eGFR between the Q1 and the Q4 were more than 30 ml/min/1.73m2. I considered that the differences were strongly associated with baseline kidney function. Therefore, it was obvious that serum creatine level was a predictor.

Answer: The authors thank to reviewer’s comment. We assumed that the differences of eGFR between Q1 and Q4 were mainly due to the wide acceptable range of normal sCr concentration. Our study results suggested that the cut-off value for prediction of CKD development was 0.75 mg/dL for females and 0.78 mg/dL for males. In the light of these cut-off values, it is worthy to subdivide normal sCr values which may give a help for risk stratification of kidney disease. For instance, hypertension was previously subclassified with normal, prehypertension, and hypertension  [1]. Likewise, sCr levels also need to be classified into normal, high-normal, and abnormal and clinicians should pay particular attention to those with high-normal sCr as well as abnormal sCr.       

Because eGFR was calculated by sCr based equation (eGFR-EPI), it is obvious that higher sCr groups showed lower eGFR levels. However, we pursued that sCr value within normal range can predict future kidney function decline and stratify the risk for incident CKD according to its level. Furthermore, in clinical field, the subject with normal sCr level as well as normal eGFR level is considered to be healthy and receives no further exams. Thus, subjects with eGFR 71.1 or those with 107.0 mL/min/1.73 m2 are treated as same as normal, and no further action is performed upon both of them. However, from the results of our study, the risk for incident CKD was apparently different between subjects with eGFR 71.1 and 107.0 mL/min/1.73 m2. This indicates that subjects with higher sCr levels should take further cares even in normal sCr ranges. We discussed this in the Discussion of the manuscript and further augmented in the revised manuscript.

Reference:

1.          Chobanian, A.V.; Bakris, G.L.; Black, H.R.; Cushman, W.C.; Green, L.A.; Izzo, J.L., Jr.; Jones, D.W.; Materson, B.J.; Oparil, S.; Wright, J.T., Jr. et al. The seventh report of the joint national committee on prevention, detection, evaluation, and treatment of high blood pressure: The jnc 7 report. Jama 2003, 289, 2560-2572.

Comment 2: Why did author add eGFR or BUN to model in analysis for outcomes? The author should conduct analysis stratified by serum creatinine levels.

Answer: The authors thank your comment. We wonder that the analysis was conducted by groups stratified by sCr levels. In addition, we did not included eGFR or BUN in the multivariable Cox models. eGFR and BUN were presented only in the Baseline characteristics (Table 1.) according to different sCr groups. If there is any miss-understanding about your comment, please let us know.

Comment 3: From previous reports, some variables such as comorbid of diabetes and hypertension are related to renal outcomes including halving of GFR. In the present study, what were the valuables associated with outcomes?

Answer: The authors appreciate the reviewer’s thoughtful comments. We performed univariable Cox analysis with all clinical parameters which were included in the multivariable Cox models (Table 2). As a result, age, BMI, SBP, history of HTN and DM, hemoglobin, fasting plasma glucose, total cholesterol, and presence of proteinuria were significantly associated with increased risk of incident CKD, whereas alcohol status and serum albumin were related to decreased risk in female subjects. Moreover, age, muscle mass, BMI, SBP, history of HTN, DM and CVDs, fasting plasma glucose, and presence of proteinuria were significantly associated with increased risk of incident CKD, whereas alcohol status, hemoglobin, and serum albumin were related to decreased risk in male subjects (Supplemental Table 2). We added this in the Result section of the revised manuscript.

 Comment 4: The prevalence of diabetes was higher in the Q1 group, on the other hand, that of hypertension was lower in the Q1 group. How did the author consider the reason?

Answer: The authors thank the reviewer’s thoughtful comment. First, the prevalence of diabetes was higher in Q1 group with statistical significance only in male group. This can be explained by age-related onset of diabetes that the mean age of Q1 group was significantly higher than others  [2]. However, we assumed that the proportion of diabetes was relatively small that this higher prevalence of diabetes with Q1 group could not have effects on the risk for incident CKD. On the other hand, the prevalence of hypertension was higher in Q4 group both in female and male subjects. Interestingly, the prevalence of proteinuria was also higher in Q4 group, which indicates that subjects with normal but higher sCr levels already have early target organ damages and endothelial dysfunction compared to those with lower sCr levels even though they were in younger age. Consequently, these early damages might have played a role for loss of kidney function  [3]. We added this in the Discussion section of the revised manuscript.

Reference:

2.          He, L.; Tuomilehto, J.; Qiao, Q.; Soderberg, S.; Daimon, M.; Chambers, J.; Pitkaniemi, J. Impact of classical risk factors of type 2 diabetes among asian indian, chinese and japanese populations. Diabetes & metabolism 2015, 41, 401-409.

3.          Drawz, P.; Rahman, M. Chronic kidney disease. Annals of internal medicine 2015, 162, Itc1-16.

 Comment 5: The author dealt serum creatine with marker of filtration. If so, creatinine clearance should be added to analysis.

Answer: The authors totally agree with the reviewer’s comment. Unfortunately, due to the prospective design of the study data (KoGES cohort study), it was impossible to measure creatinine clearance. Nevertheless, it is worthwhile to compare the predictive value of sCr value with creatinine clearance in the future study. We added this in the Limitation section of the revised manuscript.

Round  2

Reviewer 2 Report

Authors have addressed some of the reviewer's concerns. However, there have still been issues regarding the original comment 1 and 2.

Comment 1: Authors explained the importance of higher Cr levels and the fact that they used proteinuria and cardiovascular disease as additional outcomes independent of Cr levels. However, the term "CKD" has confused the reviewer in this study. As primary endpoint, authors used "incident CKD" defined by eGFR<60. In addition, authors used development of proteinuria as secondary outcome. In general, development of proteinuria means incident CKD independent of eGFR. Wether cardiovascular events were mainly ascribed to reduced eGFR or increased proteinuria remained unknown in this study. Therefore, the reviewer recommends that authors should change the descriprion of primary endpoint from "incident CKD" to "eGFR <60".

Comment 2: Authors presented additional data using MDRD equation. However, MDRD again provides estimated but not measured GFR. In addition, MDRD has poor performance to estimate GFR >60 compared with CKD-EPI. The reviewer recommends that authors should describe in detail the reason why they could not utilize measured GFR and why the validity of results in this study could be ensured by eGFR in Discussion, and data using MDRD equation should be removed from the revised manuscript.

Author Response

Responses to reviewers’ comments

# Reviewer 2

Comments to authors: Authors have addressed some of the reviewer's concerns. However, there have still been issues regarding the original comment 1 and 2.

Answer: The authors sincerely appreciate Reviewer’s valuable comments. We did our best to make point-by-point responses to each comment from the reviewer.

 Comment 1: Authors explained the importance of higher Cr levels and the fact that they used proteinuria and cardiovascular disease as additional outcomes independent of Cr levels. However, the term "CKD" has confused the reviewer in this study. As primary endpoint, authors used "incident CKD" defined by eGFR<60. In addition, authors used development of proteinuria as secondary outcome. In general, development of proteinuria means incident CKD independent of eGFR. Weather cardiovascular events were mainly ascribed to reduced eGFR or increased proteinuria remained unknown in this study. Therefore, the reviewer recommends that authors should change the description of primary endpoint from "incident CKD" to "eGFR <60".< span="">

Answer: The authors thank the reviewer for this comment. We totally agree with the reviewer’s comment and we change the description of primary endpoint from incident CKD to “eGFR<60 mL/min/1.73 m2” throughout the whole manuscript.

 Comment 2: Authors presented additional data using MDRD equation. However, MDRD again provides estimated but not measured GFR. In addition, MDRD has poor performance to estimate GFR >60 compared with CKD-EPI. The reviewer recommends that authors should describe in detail the reason why they could not utilize measured GFR and why the validity of results in this study could be ensured by eGFR in Discussion, and data using MDRD equation should be removed from the revised manuscript.

Answer: The authors appreciated the reviewer for this valuable comment. As recommended we removed data using MDRD equation in the revised manuscript.

Reviewer 3 Report

The authors discussed and revised according to my comments.

Author Response

Responses to reviewers’ comments

# Reviewer 3

Comments to the Authors: The authors discussed and revised according to my comments.

Answer: The authors sincerely thank to the reviewer for giving us thoughtful comments.
